# A primer on in vitro biological neural networks

**Frithjof Gressmann   Ashley Chen   Lily Hexuan Xie**
**Sarah Dowden   Nancy Amato   Lawrence Rauchwerger**

Siebel School of Computing and Data Science
University of Illinois at Urbana-Champaign
`{fg14,ac136,hexuanx2,sdowden2,namato,rwerger}@illinois.edu`

## Abstract

Recent advances in bioengineering have enabled the creation of biological neural networks in vitro, raising the prospect of novel, unconventional platforms that can leverage genuine biological computation. The technology could help unlock computing paradigms that could be faster, more powerful, and more energy efficient than the silicon-based architectures that dominate today's computing landscape. However, engineering cell cultures for computing applications presents a radical departure from digital von Neumann architectures that computer scientists have grown accustomed to and will require a rethink of the entire stack. Here, we provide a brief overview of the key technologies, principles, and challenges of this emerging interdisciplinary field. We argue that seizing on its potential will require the development of new machine learning approaches that can process the vast observable activity of neuronal cell cultures and learn to control and make sense of their neural code. Such an effort could provide a pathway for leveraging biological neural networks and contribute to our understanding of what makes biological learning in neurons so incredibly efficient, holding broader lessons for the development of next-generation AI systems.

## 1   Introduction

In recent years, two mutually influential fields, artificial intelligence (AI) and neuroscience, have witnessed revolutionary developments leading to new synergistic opportunities. The remarkable success of large-scale neural networks in machine learning (ML) has enabled the effective modeling of complex data patterns and relationships across diverse domains such as natural language processing, computer vision, and time-series analysis [6]. In bioengineering, groundbreaking work in cellular reprogramming has enabled the conversion of ordinary human cells to stem cells [48], facilitating the in vitro cultivation of brain cell cultures for study and application outside of their natural biological context [46]. These advances make possible an emerging interdisciplinary effort to develop machine learning models that learn to interact with in vitro biological neural networks.

This primer presents a brief and introductory overview of this development and its potential. Reviewing long-standing efforts in multiple disciplines, we illustrate a possible strategy to leverage in vitro "wetware" that integrates work on neural simulation, spiking neural networks, surrogate optimization, neuromorphic computing, and machine learning models. Apart from delivering "wetware" substrates for computing, this undertaking can also serve as a vehicle for gaining a deeper understanding of the mechanisms that make biological neural processing so efficient. Thus, we expect that engineered living biological systems will continue to drive progress in ML and neuroscience and ultimately contribute to alternative hardware for artificial intelligence applications.

38th Second Workshop on Machine Learning with New Compute Paradigms at NeurIPS 2024(MLNCP 2024).

# 2 Principles, technologies, and challenges

To begin, this section 2 reviews key principles, technologies, and challenges. Although our review is far from comprehensive, it provides pointers to surveys in the respective disciplines that shape this emerging field. Against this background, in Section 3, we sketch out a possible pathway to leverage machine learning techniques to learn to harness in vitro systems. We discuss open challenges and future directions the community might help address, before concluding in Section 4.

## 2.1 Biological neural networks

Given the significant success of neural networks in machine learning it can be easy to forget that *artificial* neurons are a radical simplification of their biological counterparts that originally inspired them. Biological neurons do not only encode information in a fundamentally different way through spiking temporal dynamics [44], but also leverage processes such as synaptic, homeostatic and structural plasticity [3, 60], local error propagation via dendritic computation [39, 33], or neuromodulation [12] whose complexity far exceeds those of artificial neurons. Thus, it is no surprise that improvements in artificial neural networks continue to seek inspiration from neurophysiology and neuroscience [32, 65]. At the same time, despite significant progress, our understanding of biological neural computing leaves much to be desired. In particular, while the huge divergence in power requirements between artificial and biological neural networks suggests a greater energy efficiency of biological neural learning, exactly how it is achieved remains unclear. At a lower level, however, the general physiological principles that give rise to the complex neural dynamics have been uncovered. Put simply, biological neurons transmit signals in the form of potential differences between ions that are separated by the cell membrane [17]. The opening of ion channels in the membrane causes the cell to depolarize, a process that propagates along the membrane toward downstream cells. Upon reaching the synaptic terminal, neurotransmitters are released that diffuse to and induce a current in the post-synaptic neuron, which continues the signal transmission chain.

## 2.2 Neural recording and stimulation

Fundamentally, these physiological processes can be manipulated in multiple ways [59]. For one, changing the extracellular potential can illicit depolarization and thus induce spiking activity. Alternatively, manipulation of the ion-channel permeability can alter the current flow and neural processing as a result. More fundamentally, neurotransmitter and blocker agents can interfere with the chemical balance at the synapses and modulate the synaptic neuro-transmission. In practice, experimental techniques leverage these principles to establish some level of control over neural dynamics.

**Multi-electrode arrays**   As one of the most established neuromodulation techniques [40, 50], electrical stimulation is commonly realized with extra-cellular electrodes that can detect and deliver potential differences in surrounding cells [43]. In particular, multi-electrode arrays (MEAs) that arrange electrodes in configurable mesh-like layouts allow high-resolution electrophysiological measurements with minimal disruption to cell tissues [10]. However, a major limitation of electrical stimulation is its inability to target specific cells and regions due to current spread [54].

**Optogenetic stimulation**   Optogenetics has emerged as a promising alternative for neurostimulation, as it uses light to manipulate specific neurons and neuron groups [15, 57, 11]. The method involves introducing foreign light-sensitive transmembrane proteins, known as opsins, into target cell populations [35]. Opsins may, for instance, be delivered via viral infection, allowing the targeting of specific cells [63]. Subsequent light stimulation can then precisely activate or deactivate ion channels and neuronal activity without affecting neighboring cells. In particular, there is a wide variety of different microbial and genetically modified opsins that allow flexible experimental design and trade-offs [34]. For example, certain opsins may respond to low-intensity light, minimizing potential cell damage [42]. Optogenetic stimulation also works well with MEA-based systems, allowing for increasingly integrated experimentation platforms [45, 8, 53]. Thus, it is no surprise that optogenetic stimulation is seeing widespread adoption for in vitro experimentation [64, 36, 23, 67, 57, 11].

**Continued development**   Besides these mainstream techniques, approaches such as magnetic stimulation [59, 9] or ultrasonic stimulation [28] continue to be developed and may emerge as additional options in the future.

### 2.2.1  Neural coding and data processing

Decoding information from biological neurons is an essential yet challenging process. Modern recording devices, such as MEAs, allow the simultaneous recording of activity from hundreds to thousands of neurons [25] making neural data extremely high dimensional. Neural processes are also inherently stochastic. Synaptic vesicles, for example, are known to spontaneously release neurotransmitters even in the absence of evoked activity, causing random activity fluctuations as a result [2]. At the same time, recording devices and techniques introduce additional noise and uncertainty. For instance, since MEAs typically record extracellularly from a bunch of cells, complex post-processing algorithms that determine which neurons fired are required, adding another layer of uncertainty [21].

Several neural coding strategies have been proposed, such as rate, temporal, rank, and direct coding, to extract and represent the information content of neural activity [47, 62]. However, it remains unclear to what degree biological neural networks actually employ such encoding schemes. Although there is significant evidence that cognitive processes depend on the precise timing of neuronal activity [52], how information processing is reliably sustained in the presence of noise is an open question.

An alternative, higher-level approach to processing and decoding neural activity involves uncovering low-dimensional representations of high-dimensional neural data. Recent work suggests that neural activity can be effectively represented in fewer dimensions, indicating that the high-dimensional nature of current neural data might be highly redundant [26]. However, identifying these low-dimensional representations remains a challenge. Neural population activity exhibits inherent nonlinearity [19], yet many widely used dimensionality reduction techniques, such as Principal Component Analysis (PCA), make linear assumptions and may not capture data patterns effectively. At the same time, non-linear dimensionality reduction methods, such as autoencoders, often struggle with issues such as noise and overfitting in neural data [1].

As such, further advances are necessary in processing and decoding neural recordings. For example, the study of neural dynamics could help fill knowledge gaps and automatically uncover functional and structural properties of neural systems [41].

### 2.3  In vitro neural networks

Notably, in vitro systems can increasingly support these efforts by providing a test bed for experimentation with realistic neural activity and extensive options for causal intervention. While the field was historically restricted to recording and stimulation technology in vivo [18], advances in bioengineering are opening up new possibilities for the study and use of neural systems and technologies in vitro.

**Induced pluripotent stem cells**   The key to this development is induced pluripotent stem cells (iPS). They are a type of pluripotent cell derived from adult somatic cells that have been reprogrammed to an embryonic-like state, providing a virtually unlimited and less ethically problematic source of cells for biomedical research [48]. Initially developed using mouse fibroblasts, the development of human iPS cells (hiPSCs) shortly followed, with tremendous implications for biomedical and adjacent fields [61, 48, 29, 37]. In particular, the cheaper and more reliable production of cells with characteristics of embryonic stem cells makes it possible to consider the development of computing applications based on living neurons [46].

**Organoids**   Notably, the so-called "organoid" technology is driving further progress to enable increasingly sophisticated applications [68]. Organoids are stem cell-derived, artificially generated three-dimensional (3D) cultures of cells. They can contain different cell types that self-organize through cell-sorting processes and spatial restrictions. Importantly, organoids can be generated in vitro from iPS cells. Researchers often opt for 3D cultures (organoids) over two-dimensional cultures (iPS cells) to obtain more physiologically realistic cellular compositions and achieve extensive culture growth, all while maintaining the potential for high-throughput screenings and analysis [51]. For instance, high-content imaging (HCI) and machine learning strategies allow for a fast analysis of data derived from organoids [13]. Overall, the technology has matured to the point where leveraging of cell cultures for computing applications is moving into the realm of distinct possibility [46].

# 3 Machine learning for leveraging in vitro neural networks

Despite the many advances and increasing technological sophistication, the engineering of cell cultures for computing applications faces, as we will argue, a bootstrapping problem. To illustrate this, consider that the development of conventional von Neumann type computers was driven by a theory of computation developed before any prototype of real-world computers would emerge. The question was not how a Turing-machine-like device could compute in theory, but how to solve a host of practical challenges to realize its real-world implementation. Engineering of biological tissue for computing, on the other hand, faces two problems at once: figuring out the *practical* challenges of this effort while at the same time developing a *theory* of how what has been developed works (or does not work). This has important methodological implications. Notably, data collection and algorithmic analysis of neural activity in itself may be of limited value as long as a formal framework for its interpretation is lacking [30]. Moreover, it is important to keep in mind that the engineered cell culture may not behave as it would in a healthy in vivo subject and is in this sense "functioning correctly". Furthermore, the high data dimensionality of recording and stimulation devices paired with the low signal-to-noise ratio makes systematic interactions with the system challenging (see Section 2). The extensive post-processing to sort and reconstruct spiking activity from the raw data adds another layer of uncertainty in itself [21, 16].

## 3.1 Towards end-to-end optimization in vitro

In practice, however, despite the numerous challenges, there is a growing list of successes in learning to control and leverage neural systems in computing applications. For instance, brain-computer interfaces that are tested with human patients have been demonstrated to decode thought from neural activity recordings with remarkable accuracy [20]. For simpler organisms like the nematode Caenorhabditis elegans, optogenetic stimulation has been used to induce basic motor control [31]. Furthermore, the activity feedback of in vitro neurons has been used to realize basic video game play [27]. Arguably, the key to these successes has been effective machine learning methods that can build rich, implicit representations of the observed system dynamics. By framing the problem as a control problem amenable to optimization, it becomes possible to steer the neural activity toward desired states and dynamics despite the limited understanding and experimental control of the neural dynamics. The continued progress in machine learning supports these developments further. The rise of attention-based transformer architectures provides a scalable and effective way to build powerful representations from large-scale pre-training corpora that can be fine-tuned to specific applications [6]. To illustrate the potential of these developments and motivate further research, we sketch out a possible work in this area in the remainder of this paper. Crucially, it may be possible to sidestep bootstrapping issues by framing the bio-engineering task as a general, end-to-end optimization problem to improve the in vitro computing capabilities while uncovering the working principles of biological information processing.

**Learning control model** While the technology and capabilities of experimental systems can vary significantly (Section 2), at a basic level, controlling in vitro cell cultures comes down to figuring out a stimulation sequence in response to observed activity. Specifically, a control model needs to learn to predict appropriate *stimulation* of the available input channels at certain *times*. This may be, for instance, a sequence of times when to deliver stimulation through certain electrodes or via laser-induced optogenetic means. As such, the control model can be characterized as a mapping $f : \mathbb{R}^{N \times T} \to \mathbb{R}^{j(x) \times T'}$ that takes $N$-dimensional inputs and outputs a sequence of system stimulation times. The optimization objective is to find a set of parameters $\theta$ such that the stimulation sequence $f_\theta(x) = \vec{k}$ steers the observable neural dynamics of the biological neural network $BNN(\vec{k}) = \vec{y}$ in some desirable way. Note that this formulation does not assume anything about the internal characteristics of the neural systems. Training $f$ successfully means not only overcoming the practical challenges of controlling a noisy, complex system, but it also implies uncovering some properties of the $BNN$ that can be exploited to achieve the given objective. For instance, the model could simply use the $BNN$ as a random projection into a higher dimensional space (this would be reminiscent of reservoir computing [14]). A more sophisticated model, however, may learn to exploit more intricate properties of the $BNN$. For example, the model may leverage present plasticity by repeatedly delivering simulations to reconfigure the synaptic connectivity of the network.

**Optimization approach**    How to optimize $f$ effectively is, in general, as much of an open question as what model and training approach would be most suitable. There are, however, principles that can guide the experimentation. First, the high cost and slow pace of lab experiments means that the training of $f$ will likely rely on a pretraining scheme using synthetic data with subsequent fine-tuning on the more limited real-world data.

Notably, the long-standing developments in high-fidelity neural simulation present a rich resource for generating realistic synthetic data of neural dynamics. Thus, developing a simulation-driven pretraining corpus for a large-scale sequence model $f$ is likely a worthwhile first step. One key question in this effort will be what level of simulation fidelity is required to allow $f$ to represent relevant neural dynamics without over-fitting. Evidence from real-world data suggests that pre-trained representations may be able to bridge considerable transfer gaps. For instance, it has been demonstrated that pre-trained representations of neural activity can be general enough to transfer to different data domains, for example, between muscular electromyographic (EMG) signals to electroencephalographic (EEG) brain activity [4]. It may thus be sufficient to generate and train on synthetic data that only loosely match the lab data encountered at fine-tuning and inference time.

With suitable and sufficient data in place, the question becomes how to optimize $f_\theta$. While non-continuous spiking dynamics are not differentiable in general, work on spiking neural networks (SNNs) has brought about a wide range of applicable optimization techniques [44, 49, 66]. In particular, surrogate gradient techniques offer a straightforward way to apply backpropagation-driven training to otherwise non-differentiable spiking dynamics [38]. Moreover, for the leaky-integrate and fire neuron model, several methods [5, 7, 58] provide exact gradients and can implement event-based gradient computation within the dynamical system [56]. These advances allow for a simulation framework that integrates the power of backpropagation-based machine learning models with theoretical and experimental models of biological neural networks [41].

With approximate or exact gradients available in simulation, it becomes possible to pre-train $f$ in an end-to-end fashion using conventional gradient-descent optimization strategies. It is worth stressing that the objective in this setting is to find parameters such that $f$ *exploits* the biological neural network to minimize the loss, as opposed to minimizing the loss through $f$ directly. This is reminiscent of a teacher-student [24] or a knowledge distillation setting [22], and work in this area may provide lessons for effective training.

**Application in vitro**    Finally, with a pre-trained model $f$ as a controller, it should be possible to "train" biological neural networks in vitro. To illustrate one possibility, consider the following approach. Training data $x$ are encoded with the pre-trained $f$ in a stimulation pattern $\vec{k}$ that is fed into both the real-world lab system and the corresponding differentiable simulation. The cell culture output activity is recorded and used to compute the backward pass in the differentiable simulator with respect to $f_\theta$. The resulting gradient that updates $f_\theta$ will differ from the unknowable "true" gradient of the in vitro neural network, but it may be good enough to ensure forward progress in the iterative fine-tuning of $f$. However, while a similar optimization approach has been used successfully to estimate the gradients of other real-world physical systems [55], how to effectively estimate such "good enough" gradients of the much more complicated in vitro systems presents an important open challenge. Leveraging simulation and data from lab experiments in such a way may produce data-driven training strategies whose effectiveness can be continually refined and experimentally verified. Overall, it is plausible that converging efforts, guided by feedback from real-world experiments, will help pave the way to increasingly sophisticated computing applications in vitro.

## 4   Conclusion

We have reviewed an emerging interdisciplinary endeavor to develop the technology to harness in vitro biological neural networks for computing applications. Key to this effort are machine learning models and optimization approaches that are able to learn to effectively interact with in vitro systems. Besides the direct practical motivations, it is likely that continued progress in this field will help uncover the working principles of biological neural processing. As such, engineered living biological networks may ultimately pave the way for next-generation hardware for artificial intelligence applications, be it in silico, in vitro, or both.

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
