# OpenReview forum: "A primer on in vitro biological neural networks"
_NeurIPS.cc/2024/Workshop/MLNCP — MLNCP Poster_

### Official Review · Reviewer_z56C · 2024-09-24

**Rating:** 7
**Confidence:** 2

**Review:**

This is a nice overview paper, which describes various biological neural networks and means of recording data from them, manipulating them and training them.

I am a bit skeptical regarding the optimism for doing the backwards pass in a differentiable simulator. I suspect a biological circuit will be a lot harder to model than the mechanical, optical and electronic circuits explored in the paper you reference (paper [55]).

---

### Official Review · Reviewer_yfBM · 2024-09-26
**This work provides an overview of the technologies, principles, and challenges associated with leveraging biological neural networks as computing substrates. It provides a nice primer to the field, but I would like to have seen more discussion of the opportunities afforded to computing.**

**Rating:** 6
**Confidence:** 3

**Review:**

This work provides an overview of the technologies, principles, and challenges associated with leveraging biological neural networks as computing substrates. The key significance of this work is that it provides background information about biological neural networks that would be relevant in thinking about how to use them as computing substrates. Such a primer is valuable to the computing community, many of whom are likely to not be aware of the challenges.

Pros:

-- Provides a nice introduction to the challenges of leveraging biological neural networks.

Cons:

-- I would like to have seen more discussion about the opportunities and unique computational characteristics of using biological neural networks. The work focused more on the underlying principles and challenges than the opportunities these provide.

---

### Decision · Program_Chairs · 2024-10-10

Accept (Poster)